# Comparison of Preoperative ECG Screening and Device-Based Vector Analysis in Patients Receiving a Subcutaneous Implantable Cardioverter-Defibrillator

**DOI:** 10.3390/medicina59122186

**Published:** 2023-12-16

**Authors:** Szymon Budrejko, Agnieszka Zienciuk-Krajka, Ludmiła Daniłowicz-Szymanowicz, Maciej Kempa

**Affiliations:** Department of Cardiology and Electrotherapy, Faculty of Medicine, Medical University of Gdansk, Smoluchowskiego 17, 80-214 Gdansk, Poland; agzien@gumed.edu.pl (A.Z.-K.); ludwik@gumed.edu.pl (L.D.-S.); kempa@gumed.edu.pl (M.K.)

**Keywords:** sudden cardiac death, implantable cardioverter-defibrillator, subcutaneous implantable cardioverter-defibrillator, sensing, electrocardiographic screening

## Abstract

*Background and Objectives*: Subcutaneous implantable cardioverter-defibrillators (S-ICDs) provide protection against sudden cardiac death from outside the cardiovascular system. ECG screening is a prerequisite for implantation, but the reproducibility of its results post-operatively in the device is only partial. We aimed to compare the results of ECG screening with device-based sensing vector analysis. *Materials and Methods*: We screened the hospital records of all S-ICD recipients in our clinic. All of them had pre-operative ECG screening performed (primary, secondary, and alternate vectors). The results were compared with device-based vector analysis to determine the relation of the pre- and post-operative vector availability. *Results*: Complete ECG screening and device-based vector analysis were obtained for 103 patients. At least two acceptable vectors were found in 97.1% of the patients pre-operatively and in 96.1% post-operatively. When comparing vectors in terms of agreement (OK or FAIL) pre- and post-operatively, in 89.3% of the patients, the result for the primary vector was the same in both situations; for the secondary, it was in 84.5%, and for the alternate, it was in 74.8% of patients, respectively. In 55.3% of patients, all three vectors were labeled the same (OK or FAIL); in 37.9%, two vectors had the same result, and in 6.8%, only one vector had the same result pre- and post-operatively. The number of available vectors was the same pre- and post-operatively in 62.1% of patients, while in 15.5%, it was lower, and in 22.3% of patients, it was higher than observed during screening. *Conclusions*: Routine clinical pre-operative screening allowed for a good selection of candidates for S-ICD implantation. All patients had at least one vector available post-operatively. The final number of vectors available in the device-based analysis in most patients was at least the same (or higher) than during screening. The repeatability of the positive result for a single vector was high.

## 1. Introduction

An implantable cardioverter-defibrillator was invented to offer protection against sudden cardiac death due to ventricular arrhythmias [1]. Typical transvenous systems were found to be very effective in that field, although the lead is an Achilles heel of the transvenous system. Lead-related complications, such as lead failure and infective endocarditis, may become a limiting factor in many patients [2]. A subcutaneous implantable cardioverter-defibrillator (S-ICD) was then designed to overcome those limitations [3,4]. The S-ICD system consists of a device can and a subcutaneous lead. The can is implanted typically on the left lateral side of the chest into the intermuscular pocket located between the serratus anterior and latissimus dorsi muscles. The lead is tunneled from the pocket to the region of the xiphoid process of the sternum and then along the left margin of the sternum towards its manubrium. This design of the system was found to have the best properties in terms of defibrillation efficacy by using the location in the subcutaneous tissue of the chest [3]. The main limitation of the system is its inability to provide permanent cardiac pacing or anti-tachycardia pacing. Only the short run of post-shock pacing is available, intended to protect against asystole after high-voltage therapy, at the cost of a painful experience to the patient (as it resembles transcutaneous rather than intracardiac pacing). In the early years, S-ICDs were considered to be an alternative to transvenous systems in very specified cohorts of patients (without vascular access or with a history of complications of transvenous ICD therapy). Currently, all patients requiring an ICD may be qualified for a subcutaneous system unless they need permanent cardiac pacing due to bradycardia, cardiac resynchronization therapy, or anti-tachycardia pacing for monomorphic ventricular tachycardia [2]. Before the patient is considered eligible for an S-ICD, one important condition has to be met, and that is a positive result of pre-operative ECG screening. Such an examination is conducted with a proprietary programmer of the S-ICD producer (Boston Scientific, Marlborough, MA, USA). Three surface leads similar to the ECG’s limb leads I, II, and III are evaluated, which emulate the future sensing vectors of the system (i.e., the primary, secondary, and alternate vectors). The system has three available sensing poles: the can and two sensing rings (the so-called A-ring at the tip of the lead and the B-ring proximally on the course of the lead). The primary vector is recorded between the can and the B-ring, the secondary vector is between the can and the A-ring, and the alternate vector is between both rings. ECG patches for screening are placed on the patient’s chest in planned future anatomical locations of the sensing poles of the device. They are connected to the programmer equipped with screening software, according to the system manual and on-screen instructions. The result of screening (labeled either “OK” or “FAIL” for each assessment in each body position) is displayed on-screen and may be saved as a PDF in the medical records (Figure 1), together with all three analyzed ECG tracings (Figure 2). Apart from the final decision of the in-built algorithm, no other parameters are presented (such as the precise scoring of ECG tracings). They are only accessible to the technical support of the manufacturer, as some technical files are available for download from the programmer in case of any problems.

The positive result of screening maximizes the chance of appropriate sensing after implantation of the S-ICD system. Meticulous screening is especially important in some specific cohorts of patients (suffering from congenital heart disease, arrhythmogenic right ventricular cardiomyopathy, or some channelopathies, such as Brugada syndrome). In those patients, the specific ECG pattern or its variability in time may prove inappropriate for S-ICD sensing algorithms and result in inadequate interventions [5,6,7,8,9]. Patients with dynamic ECG patterns (transient intraventricular conduction abnormalities, intermittent bundle branch block, or transient ventricular pacing) are most problematic in those terms [10,11].

The system producer recommends that at least one vector is positive in various body positions to consider a patient eligible for an S-ICD. In some centers (including our center in Gdansk, Poland), two positive vectors in two body positions (supine and standing) are required. To further complicate the issue, vector analysis is dynamic in its nature, and may give conflicting results in repeated measurements [12]. After implantation, the S-ICD system performs an automated setup, analyzing all three sensing vectors in two body positions and selecting the best vector based on the device algorithms. Until recently, the final vector choice was the only result of that analysis available for the implanting physician. More in-depth analysis was only available for the technical services of the producer. Recently, the system software was changed. Nowadays, the device produces a vector analysis table with all three vectors in two body positions (labeled OK or FAIL), the reason for failure if that is the case, and the final automated vector choice made by the device. An exemplary result of post-operative automated vector analysis is shown in Figure 3. Similarly to the pre-operative screening, no insight into the signal analysis is available to the operator, apart from the final decision of the sensing algorithms. The best vector, according to the signal scoring by the algorithm, is automatically selected for permanent use, and it is marked in the description and is in green color in the table. That choice may be overridden by the operator with manual vector selection, but it is strongly discouraged unless significant reasons for such a change occur.

The aim of our study was to compare the results of the vector analysis during pre-operative ECG screening with the post-operative device-based vector check in order to determine the real-life value of pre-operative screening in our everyday clinical routine and the repeatability of vector evaluation. In particular, we aimed to investigate how many vectors were available post-operatively in comparison to the pre-operative screening, if they were the same vectors, and if the total number of available vectors increased or decreased in the post-operative assessment.

## 2. Materials and Methods

We included all consecutive patients who underwent an S-ICD system (Boston Scientific, Marlborough, MA, USA) implantation from 23 October 2014 to 16 October 2023 in our department (Department of Cardiology and Electrotherapy, Faculty of Medicine, Medical University of Gdansk, Gdansk, Poland), according to the contemporary clinical guidelines and national reimbursement regulations. These regulations state that S-ICD systems can be implanted in patients having indications for an ICD not requiring permanent cardiac pacing and having one of the additional reasons to choose a subcutaneous system instead of a transvenous one. These reasons include no vascular access, young patient’s age (or, in other words, long life expectancy), high risk of infection of a transvenous system, or a history of device-related infection (local or systemic). The implanting center has to be experienced in ICD implantation and transvenous lead extraction. These regulations might have biased the selection of patients for our study, as our department serves as a referral center for the whole region and collects problematic patients from other hospitals. On the other hand, the final cohort possibly represents a typical clinical practice of S-ICD treatment in our country.

All patients had pre-operative ECG screening performed, which was considered positive if at least two vectors passed in both body positions (supine and standing). In other words, if the result was not acceptable in only one body position, then the vector was considered unacceptable as a whole. In the early days of S-ICD therapy (2012–2017), screening was executed with an ECG template provided by the manufacturer and applied to the tracings of the modified ECG leads (imitating future sensing vectors). In 2017, the producer implemented dedicated software developed to analyze sensing vectors automatically. Screening may now be performed with the use of a typical device programmer. In that analysis, the label ‘OK’ is used by the programmer for the acceptable result, and the label ‘FAIL’ is used for the negative result. No further details are available, and visual inspection of the ECG tracings does not give any insight into the details underlying the final decision. It may be helpful, though, for understanding the general ECG characteristics (such as a widened QRS, intraventricular conduction disturbances, and the problematic R-to-T ratio). Precise scoring is available only to the technical staff of the manufacturer and may be acquired upon request in selected problematic cases.

If the typical lead position on the left margin of the sternum is not positive, the right-sided potential course of the lead is checked. The primary vector is registered between the device can and the sensing ring located in the region of the xiphoid process of the sternum, while the secondary vector is between the ring at the tip of the lead (14 cm above the lower ring, usually in the region of the sternomanubrial junction) and the can, and finally, the alternate vector is between the two sensing rings, without the involvement of the device can. The location of the ECG adhesive patches for screening was based on typical anatomical landmarks, as recommended by the producer in the device manual, and as used in real-life circumstances in most implanting centers. No other methods were used (such as X-ray-enhanced screening, where screening may be preceded by the precise location of future system elements with the help of a chest X-ray and a dummy S-ICD system for possibly the best adjustment to an individual’s anatomy at that stage of decision-making). We accepted one positive vector as sufficient for S-ICD implantation only in exceptional cases where a transvenous system could not be implanted and when an S-ICD was the only option for a given patient to achieve protection against sudden cardiac death (history of infection, no vascular access). Then, the follow-up clinic records were analyzed to determine the results of routine post-operative device-based vector checks. It was performed on every patient during one of the initial follow-up visits. During that analysis, the device also labels vectors as ‘OK’ or ‘FAIL’. We considered the vector acceptable if it was labeled ‘OK’ in both body positions. The results of pre- and post-operative analyses were compared to determine the ability of pre-operative screening to predict post-operative vector availability. Moreover, demographical and clinical data were extracted from the hospital and follow-up clinic records. The study design was approved by the Bioethics Committee for Scientific Research at the Medical University of Gdansk, Poland (decision number: KB/390/2023, date of approval: 7 July 2023).

Continuous variables were presented as the mean and standard deviation (if normally distributed) or the median and interquartile range (in case of non-normal distribution). Categorical parameters were presented as numbers and percentages. The normality of distribution was tested using the Shapiro–Wilk test. Statistical significance was assumed at a p-value below 0.05. Data management and statistical analysis were performed in Microsoft Excel, R version 4.1.2 (2021-11-01, “Bird Hippie”, The R Foundation for Statistical Computing, Vienna, Austria), and R-studio software (2023.09.1 build 494, Posit Software, Boston, MA, USA).

## 3. Results

S-ICD systems were implanted from 23 October 2014 to 16 October 2023 in 125 patients. Complete pre-operative ECG screening and post-operative vector analysis were available for 103 of them. The demographical and clinical data of patients are presented in Table 1 (categorical variables) and Table 2 (continuous variables).

The pre-operative screening was positive, with at least two positive vectors in 100 patients (97.1%). In three patients (2.9%), we proceeded to S-ICD implantation despite only one acceptable vector, as a transvenous ICD was not an option in those cases. In the post-operative analysis, at least two acceptable vectors were found in 99 patients (96.1%). In the remaining four patients (3.9%), there was only one positive vector. The percentages of positive results for each of the vectors (i.e., primary, secondary, and alternate) in the pre-operative and post-operative analyses are presented in Figure 4. The percentages of patients in whom one, two, or three acceptable vectors were found in both analyses are presented in Figure 5.

Comparing the vectors in terms of agreement (positive or negative) in the pre-operative screening and post-operative vector check in 92 patients (89.3%), the result for the primary vector was the same (either ‘OK’ or ‘FAIL’) in both situations, for the secondary, in 87 patients (84.5%), and for the alternate vector, in 77 patients (74.8%), respectively.

In 57 patients (55.3%), all three vectors were labeled the same (either ‘OK’ or ‘FAIL’) pre- and post-operatively by the algorithms; in 39 patients (37.9%), two vectors were unchanged, and in 7 patients (6.8%) only one vector had the same result pre- and post-operatively.

The number of available vectors was the same in the ECG screening and device-based vector analysis in 64 patients (62.1%), while in 16 patients (15.5%) fewer vectors were available after implantation than in the ECG screening, but interestingly, in 23 patients (22.3%), the final number of available vectors was higher than observed during the screening. In Table 3, we present the relation of the total number of positive vectors in the ECG screening and post-operative analysis in all patients.

After the device-based vector check, one of the vectors (considered the best by the automated algorithm) was selected for permanent use. In our cohort, the primary vector was selected in 54 patients (52.4%), the secondary vector in 30 (29.1%), and the alternate in 19 patients (18.4%), respectively.

## 4. Discussion

A subcutaneous implantable cardioverter-defibrillator may be used as an alternative to the transvenous device in patients not requiring cardiac pacing, anti-tachycardia pacing, or resynchronization therapy [2]. Positive pre-operative ECG screening is a prerequisite for qualification for S-ICD implantation; however, it fails in 7–15% of unselected potential recipients [13,14,15]. However, temporary changes may occur in ECG tracings, resulting from diurnal variability [16] or transient abnormalities, such as ischemia [17]. Even if minimal, they may influence a patient’s eligibility for an S-ICD. As a result of all those phenomena, the repeatability of screening is not complete, and a more thorough analysis (possibly with repeated recordings or specialist analysis by the technical support of the company) may be needed in some borderline cases. In some clinical circumstances, ECG abnormalities may unpredictably fluctuate in long-term observations or even become permanent, which hinders the use of an S-ICD. Such an issue may occur in congenital heart disease [5,6,7,18], channelopathies [8,9], hypertrophic cardiomyopathy [19], or in patients with a paced rhythm [10,11,20]. In some patients, specific maneuvers are needed to improve the exclusion of truly ineligible candidates (ajmaline test in Brugada syndrome; exercise test in Brugada syndrome, hypertrophic cardiomyopathy (HCM), or ARVC) [9,21,22,23]. On the contrary, patients’ eligibility may be enhanced by other maneuvers, such as screening on the right side of the sternum (instead of the left side) [24,25,26,27] or X-ray-enhanced screening in the case of complex and atypical anatomy [28].

Positive screening is intended to decrease the risk of inappropriate sensing and shocks (IAS). The IAS rate in the early years of S-ICD therapy was relatively high [29]; however, with improved sensing algorithms, such as Smart Pass [30,31], it was later decreased to values comparable to TV-ICDs [32,33,34,35,36,37].

There were some attempts to substitute screening with an analysis of conventional ECG, even with the use of machine learning or artificial intelligence [6,37,38,39]. Nonetheless, the typical screening procedure is still the only one recommended by the producer of the system. The open issue remains, however, about what is the number of passing vectors needed to consider the overall result of screening acceptable. The official recommendation is one vector. In our center, we prefer to have at least two. According to some research, a value of less than three is the risk factor for IAS [40]. On the other hand, if indications to choose a subcutaneous system are strong (i.e., if a transvenous system is not an option due to anatomical abnormalities or no vascular access), then screening could be double- or triple-checked, possibly with slight adjustments of the planned position of the system components. Repetitive assessments may sometimes produce varying results. Such suspicious tracings may be, in fact, borderline acceptable and eventually positive after some fine-tuning.

There is some evidence that vectors remain stable along patients’ growth [41]. However, in some specific clinical conditions, they may change, or the device sensing vector setting may need to be changed. Those circumstances include IAS in the course of air entrapment in the pocket [42], device exchange due to battery depletion [43], or the so-called sense-B noise, a novel and hitherto unexplained sensing abnormality specific to S-ICD systems (where reprogramming to the secondary vector may be a non-surgical solution) [44,45,46]. There is also a report that the alternate vector may be less vulnerable to the oversensing of myopotentials [47]; however, this phenomenon may be related to the type of exercise [48]. Therefore, having a choice of vectors for permanent use seems reasonable in the long run.

In our study, the protocol of pre-operative screening adopted in our institution guaranteed that none of the patients had less than one acceptable vector post-operatively, and most patients had at least two (96.1%).

The overall number of available vectors was the same in screening and post-operatively in 62.1% of patients. In 15.5%, fewer vectors were available after implantation than in screening, but in 22.3%, the final number of vectors available for permanent use was higher than observed during screening. Despite the total number being acceptable, the vectors available post-operatively were not always precisely the same as those observed in the screening. In 55.3% of patients, all three vectors were in agreement (pre- and post-operatively); in 37.9% of patients, two vectors were unchanged, and in 6.8%, only one vector had the same result pre- and post-operatively. The highest repeatability of the result (either ‘OK’ or ‘FAIL’) was observed for the primary vector (89.3% of patients), followed by the secondary (84.5%) and the alternate (74.8%). It means that the overall reliability of screening is not complete, and some changes with respect to the baseline screening have to be expected in the device-based analysis. To add more complexity, the device-based analysis also has its own variability: vectors themselves, as well as an automated vector choice, may change during the follow-up [43].

In a study of 33 patients with already-implanted S-ICDs, Bögeholz et al. compared the sensing vectors recorded by the device with an ECG screening using manual and automated screening tools (as soon as the latter was introduced into clinical practice) [49]. The eligibility of sensing vectors in the device was assessed visually and established if consecutive QRS complexes were appropriately annotated on the live screen without gaps or double counting. Unfortunately, they could not report the device decision about vector acceptance or failure because it was not available or displayed by the programmer at that time. Therefore, we cannot compare our findings with that study. Nonetheless, there were some patients who had functionally acceptable sensing (as defined above) in the same vectors that had negative results in the screening, which is somewhat concordant with our observations.

Deliberate implantation with no acceptable vectors in pre-operative screening has also been described with adequate post-operative sensing. It was either preceded by modified screening with an external S-ICD device soldered to the ECG leads in the era of the template-based screening [50] or with no additional evaluation (nowadays) when screening is automated and imitates the device algorithms [51]. In such extreme situations (where the S-ICD is the only available choice), an individualized approach has to be the rule. Implantation of a subcutaneous system in such patients is always associated with a risk of inappropriate sensing and, thus with the inability of the device to operate, meaning also that the system may require explantation. On the contrary, in the study mentioned above [49], 6.1% of patients with an S-ICD system already implanted, without any sensing disturbances, were considered ineligible according to the programmer-based automated ECG analysis (i.e., screening-like but in patients already having S-ICDs).

The fact that vectors are available for permanent use may be significant (e.g., the secondary vector to be used in case of sense-B noise). As mentioned above, there is no guarantee that the pre-operative observations will hold true after a device’s implantation. Nonetheless, there is no official recommendation regarding the specific vector selection, rather, only the number of vectors (at least one, according to the producer).

Device-based algorithms were described as not being completely concordant with ECG screenings in one study [52]; however, the authors of that study applied algorithms in Matlab software to the external ECG recordings and compared them with the manual ECG screening. Another interesting approach was reported by Brouwer et al. regarding the correlation of screening with device algorithms [50]. Patients considered non-eligible with standard screening (with a manual template) were then re-screened using an external S-ICD system with sensing poles connected mechanically to the adhesive ECG patches and using the sensing algorithms of the device itself. Six out of eight patients had a positive result of such modified screening and were implanted with good results. Those results are not directly comparable with ours, as an ECG screening was exclusively template-based in that study. The authors do not report vectors derived from the device, as the study was performed before the era when the results became directly available to the programmer. Moreover, even the precise device algorithm applied to the ECG from the skin surface is not exactly the same as the device-based analysis after implantation. Since 2017 (and with a technical update in 2021), the programmer has performed ECG screenings with the same sensing algorithm as the implanted device. Potential differences and discrepancies between the screening and device-based analysis seem to result merely from the different points of signal acquisition (the surface of the skin in screening and the deep subcutaneous layer post-implantation); thus, they cannot be predicted with 100% precision. Additionally, the Smart Pass algorithm, enhancing detection and sensing, is applied by the device only after an automated signal analysis. All the above phenomena leave us with some degree of uncertainty regarding the final outcome of vector configuration and sensing.

It can be anticipated that screening has to be performed with extreme precision regarding an ECG’s lead placement to allow for as high an approximation of the final location of the device components as possible. It is even more important if only one vector is acceptable, with an option to use special methods, such as an X-ray-enhanced approach in complex cases.

Screening is mandatory, and it is highly reproducible in sensing vectors after implantation. Nonetheless, some minor variations are possible. Fewer vectors may be available for permanent use than expected after screening, which leaves room for improvement in the device’s sensing algorithm. An ideal scenario would incorporate playing with the axis of the QRS complex using all three available vectors and computing an optimal signal. Another improvement might be to allow the device to change the permanently selected vector on the go if the QRS morphology changed and the algorithm decided that there was a better option available. Unfortunately, such optimizations would require re-certification of the whole system and software, which seems unlikely. On the other hand, the number of available vectors after implantation may be higher than expected. In extreme cases, it allows for implantation in patients with completely failed screenings. Successful attempts of that type have already been described, although it may be considered a form of gambling, with the cost of the device at stake. In case of failed sensing, the system would have to be explanted and replaced with a transvenous one. Although we do not have any insights into the possible future sensing improvements, some progress in S-ICD therapy is on the horizon. The manufacturer is developing a system with a combined leadless pacemaker, possibly expanding the group of potential recipients. With the advent of a competitive extra-vascular ICD by Medtronic into clinical practice (with a lead implanted under the sternum and anti-tachycardia pacing available), the future of extracardiac ICDs seems promising.

## 5. Limitations of the Study

Our analysis is a retrospective study. The pre-operative screening was based on anatomical landmarks and cannot be fully correlated with the final position of the system components. Therefore, an unknown part of the vector variability may be due to a mismatch of the anatomical position between the pre-operative ECG lead location and the post-operative position of the system components. Nonetheless, the study aimed to evaluate the real-life conditions and everyday clinical practice in this field, according to the producer’s recommendations.

## 6. Conclusions

In our cohort of patients, routine clinical pre-operative screening allowed for a good selection of candidates for S-ICD implantation. All patients had at least one vector, and 96.1% of patients had at least two vectors available post-operatively. The final number of vectors available in the device-based analysis was lower than during screening in 15.5% of patients, but in the remaining ones, it was at least the same or higher. The pre- and post-operative repeatability of the positive results for single vectors was high, ranging from 74.8% for the alternate, through 84.5% for the secondary, and reaching 89.3% for the primary vector, respectively.

## Figures and Tables

**Figure 1 medicina-59-02186-f001:**
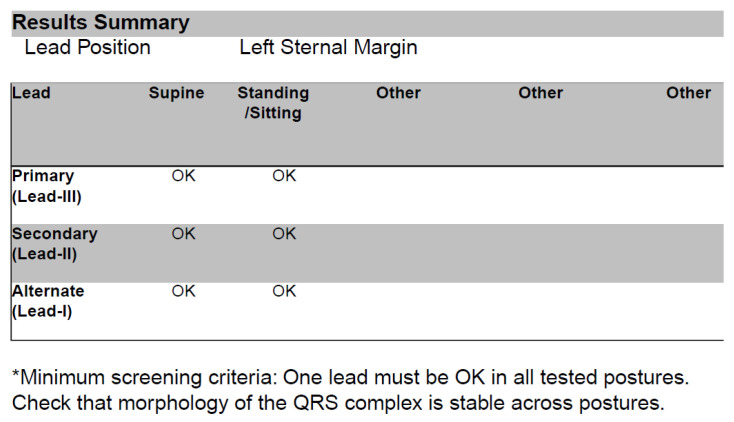
Summary of pre-operative screening in a candidate for implantation of a subcutaneous cardioverter-defibrillator. Two body positions were investigated: supine and standing.

**Figure 2 medicina-59-02186-f002:**
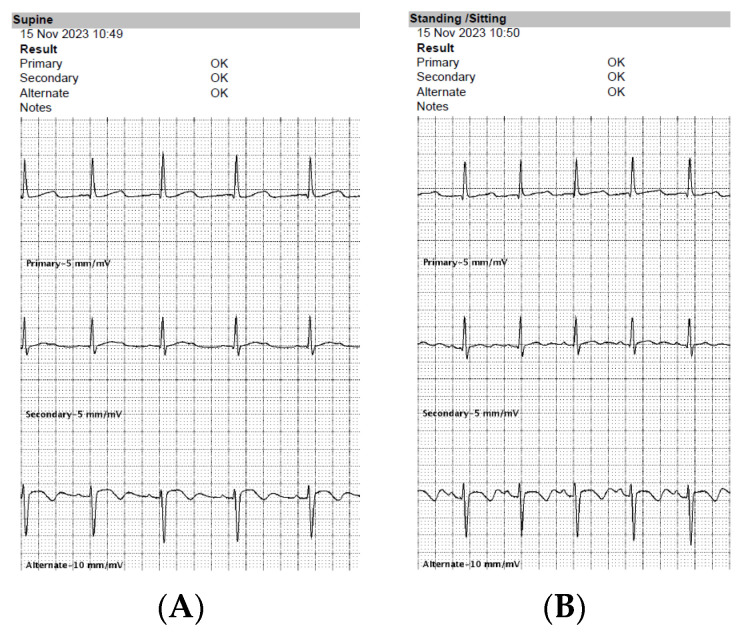
Details of pre-operative screening in a candidate for implantation of a subcutaneous cardioverter-defibrillator. ECG tracings and screening results for two body positions: supine (panel **A**) and standing (panel **B**).

**Figure 3 medicina-59-02186-f003:**
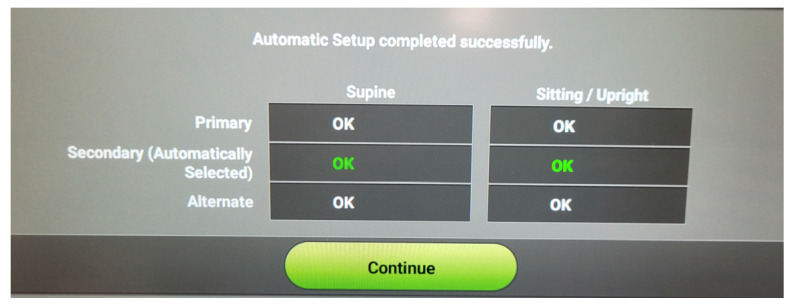
Exemplary result of post-operative vector analysis. Three vectors were analyzed in two body positions according to the protocol of the automated setup. The vector selected for permanent use is marked in green by the device.

**Figure 4 medicina-59-02186-f004:**
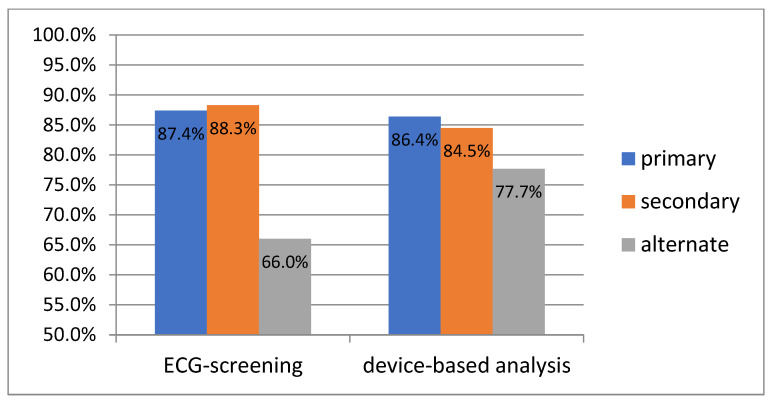
The percentages of positive results for each of the vectors (i.e., primary, secondary, and alternate) in the pre-operative ECG screening and post-operative device-based vector analysis.

**Figure 5 medicina-59-02186-f005:**
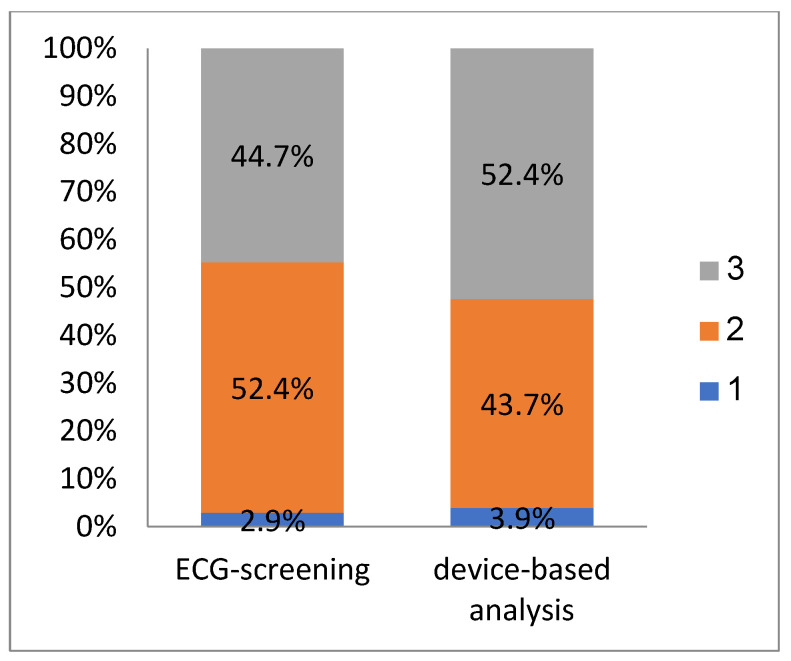
Percentage of patients with 1, 2, or 3 positive vectors in the pre-operative ECG screening and post-operative device-based vector analysis.

**Table 1 medicina-59-02186-t001:** Demographical and clinical data, categorical variables.

Variable	Number (%)
Male sex	72 (70)
**Indications**	
ICM	31 (30.1)
NICM	30 (29.1)
LQTS	4 (3.9)
LVNC	2 (1.9)
IVF	22 (21.4)
ARVC	3 (2.9)
HCM	5 (4.9)
other	6 (5.8)
Primary prevention indication	52 (50.5)
**Clinical history**	
HA	32 (31.1)
DM	15 (14.6)
CKD	5 (4.9)
CABG	2 (1.9)
PCI	23 (22.3)
Prior sternotomy	8 (7.8)
Prior CIED	30 (29.1)
Prior TV-ICD	30 (29.1)
Prior TLE of TV-ICD	29 (28.2)
**NYHA**	
I	45 (43.7)
II	50 (48.5)
III	8 (7.8)
**Cardiac rhythm**	
SR	98 (95.1)
AF	5 (4.9)
**Indication for S-ICD (more than one factor per patient possible) preference over TV-ICD**	
No vascular access	19 (18.4)
Increased risk of infection	13 (12.6)
Young age	82 (79.6)
CDRI	7 (6.8)
CDRIE	7 (6.8)
**Implantation technique**	
2-incision/3-incision	66 (64.1)/37 (35.9)
Subcutaneous/intermuscular pocket	1 (1)/102 (99)
Lead on the left/right sternal margin	101 (98.1)/2 (1.9%)

AF—atrial fibrillation; ARVC—arrhythmogenic right ventricular cardiomyopathy; CABG—coronary artery bypass graft; CDRI—cardiac device-related infection; CDRIE—cardiac device-related infective endocarditis; CIED—cardiac implantable electronic device; CKD—chronic kidney disease; DM—diabetes mellitus; HA—hypertension; HCM—hypertrophic cardiomyopathy; ICM—ischemic cardiomyopathy; IVF—idiopathic ventricular fibrillation; LVNC—left ventricle non-compaction; LQTS—long QT syndrome; NICM—non-ischemic cardiomyopathy; NYHA—New York Heart Failure; PCI—percutaneous coronary intervention; S-ICD—subcutaneous implantable cardioverter-defibrillator; SR—sinus rhythm; TLE—transvenous lead extraction; TV-ICD—transvenous implantable cardioverter-defibrillator.

**Table 2 medicina-59-02186-t002:** Demographical and clinical data, continuous variables.

Variable	Range	Median	1st–3rd Quartile
Age (years)	13–82	45	34–56.5
Height (cm)	156–197	176	169–175.3
Weight (kg)	49–130	80	70–92
BMI (kg/m^2^)	17–37	26	22–29
LVEF (%)	10–75	35	25.5–56
ECG—RR (ms)	620–1500	860	769.5–1000
ECG—PQ (ms)	96–285	171	160–185.5
ECG—QRS (ms)	60–186	103	94.5–114
ECG—QT (ms)	320–680	402	389–440

BMI—body mass index; ECG—electrocardiogram; LVEF—left ventricle ejection fraction; PQ—PQ interval; QRS—duration of the QRS complex; QT—QT interval; RR—RR interval.

**Table 3 medicina-59-02186-t003:** Comparison of the relation of the number of passing vectors in pre-operative ECG screening and post-operative device-based analysis in subsequent patients.

		Pre-Operative Number of Acceptable Vectors		
**Post-operative number of acceptable vectors**		**3 Vectors**	**2 Vectors**	**1 Vector**
	**3 vectors**	32 patients	22 patients	0 patients
	**2 vectors**	14 patients	30 patients	1 patient
	**1 vector**	0 patients	2 patients	2 patients

## Data Availability

The data presented in this study are available on request from the corresponding author. The data are not publicly available due to the policy of our hospital and the restrictive internal procedure of acquisition of data, even if anonymized.

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
