# Peer review of "Comparison of Preoperative ECG Screening and Device-Based Vector Analysis in Patients Receiving a Subcutaneous Implantable Cardioverter-Defibrillator"

_medicina, 2023, doi:10.3390/medicina59122186_

Round 1

Reviewer 1 Report

Comments and Suggestions for Authors

Manuscript Title: Comparison of preoperative ECG screening and device-based vector analysis in patients receiving a subcutaneous implantable cardioverter-defibrillator

Authors: Szymon Budrejko et al.

Journal: Medicina (2023)

Review Date: November 13th, 2023

1. Summary:

The manuscript presents a comparison of the results from preoperative ECG screening and post-operative device-based vector analysis in patients receiving subcutaneous implantable cardioverter-defibrillators (S-ICDs). The study's objective is to ascertain the reliability of preoperative ECG screening in predicting postoperative vector availability and the repeatability of vector evaluations. The results indicate that routine preoperative screening is an effective measure for the selection of suitable candidates for S-ICD implantation, with a high repeatability rate for positive vector results.

2. Originality / Novelty:

The study's approach to comparing preoperative ECG screening with post-operative device-based analysis in the context of S-ICD implantation provides incremental knowledge to the cardiology field. While the concept of ECG screening is established, the manuscript offers new insights into the effectiveness and reliability of preoperative assessments.

3. Significance of Content:

The research findings are significant as they may influence clinical practice by reinforcing the value of preoperative ECG screening and potentially improving patient outcomes in S-ICD implantations.

4. Quality of Presentation:

The manuscript is well-organized and the data is presented clearly through tables and figures. However, the manuscript would benefit from additional details in the methods section regarding patient selection and exclusion criteria, and a more thorough discussion on how this study adds to the current body of literature.

5. Scientific Soundness:

The methodology is sound, with clear descriptions of the ECG screening process, device-based vector analysis, and statistical methods employed. The study design is appropriate for the research question, and the conclusions drawn are supported by the results.

6. Interest to the Readers:

The study addresses a focused question that is of particular interest to clinicians involved in the implantation and management of S-ICD systems. Its relevance might be less pronounced for a broader readership.

7. Overall Merit:

The manuscript provides valuable contributions to the field of cardiology, particularly in the domain of S-ICD implantation. The study is methodologically robust, and its findings have the potential to influence clinical practice.

8. Recommendations:

  • Minor Revision:
    • The authors should provide a more detailed description of patient selection criteria ( espec. national reimbursement regulations  - could it bring some selection bias?) to reinforce the study's validity.
    • Clarification on the methods used for statistical analysis and any measures taken to control for potential confounders or biases is recommended.
    • Did the authors consider analysis of factors that were associated with the admittedly rare ( according to the authors) but nonetheless described "failure " to match preoperative ECG screening and device-based 2 vector analysis ?
    • A thorough proofreading is advised to correct minor grammatical errors and improve readability – long senteces, long paragraphs
    • Expansion of the discussion on the potential clinical implications of the study would be advantageous - Could the authors define a "potential recommendation" for S-ICD producers of vector screeing method ?

9. Conclusion:

Overall, the study is well-executed with findings that are likely to have a positive impact on clinical practices concerning S-ICD implantation. The manuscript would benefit from revisions that clarify methodological details and contextualize its findings within the broader research landscape.

Comments on the Quality of English Language

all mentioned above

Author Response

We would like to thank Reviewer #1 for the revision of our article and all the comments.

Comment: The manuscript is well-organized and the data is presented clearly through tables and figures. However, the manuscript would benefit from additional details in the methods section regarding patient selection and exclusion criteria, and a more thorough discussion on how this study adds to the current body of literature.

Authors’ response: We added more information about patient selection and methods. The discussion has also been expanded.

Comment: The authors should provide a more detailed description of patient selection criteria ( espec. national reimbursement regulations  - could it bring some selection bias?) to reinforce the study's validity.

Authors’ response: The requested information has been added.

Comment: Clarification on the methods used for statistical analysis and any measures taken to control for potential confounders or biases is recommended.

Authors’ response: Only descriptive statistical methods were required for the analysis, except for the normality test (Shapiro-Wilk, as stated in the manuscript). Information about potential selection bias has been added to the text.

Comment: Did the authors consider analysis of factors that were associated with the admittedly rare (according to the authors) but nonetheless described "failure " to match preoperative ECG screening and device-based 2 vector analysis ?

Authors’ response: Factors for screening failure are known to some extent, and include congenital heart disease, budle branch block or intraventricular conduction abnormalities, as well as abnormalities of repolarization (for example in HCM, Brugada syndrome or ARVC). This has been mentioned in the introduction and discussion. We did not analyze that phenomenon, as it would require gathering information about all failed screenings and the possible underlying causes. Unfortunately, we do not have such data available. Our goal was to compare real-life positive screening qualifying for S-ICD implantation and subsequent device-based sensing analysis, as it has some potential implications for patient selection.

Comment: A thorough proofreading is advised to correct minor grammatical errors and improve readability – long senteces, long paragraphs

Authors’ response: Additional proofreading was performed for suggested improvements.

Comment: Expansion of the discussion on the potential clinical implications of the study would be advantageous - Could the authors define a "potential recommendation" for S-ICD producers of vector screening method ?

Authors’ response: We added our comments regarding that issue to the text.

Reviewer 2 Report

Comments and Suggestions for Authors

The background provides a clear context for the study, emphasizing the importance of S-ICDs in preventing sudden cardiac death. However, it lacks some details about the significance of reproducibility and its impact on patient outcomes.

            The objectives of the study are well-defined, aiming to compare preoperative ECG screening results with post-operative device-based vector analysis. This objective addresses a relevant clinical concern and sets the stage for the research.

            The materials and methods section provides a concise overview of the study design and data collection. However, it lacks information about the specific criteria for "acceptable vectors" and the methodology for device-based vector analysis. Adding more details about these aspects would enhance the transparency of the study.

            The results are presented clearly, with percentages and comparisons provided to highlight key findings. The high agreement between pre- and post-operative vectors for the primary, secondary, and alternate vectors is a significant positive outcome, indicating good repeatability.

            The conclusions drawn from the study are straightforward and well-supported by the results. It is emphasized that routine preoperative screening is effective in selecting suitable candidates for S-ICD implantation. The information about the availability and repeatability of vectors post-operatively is a key takeaway.

Overall, the study addresses an important clinical issue and provides valuable insights. To enhance the paper, consider discussing the clinical implications of the findings in greater depth. Additionally, a brief discussion of future research directions would be beneficial.

Comments on the Quality of English Language

Minor editing of English language required.

Author Response

We would like to thank Reviewer #1 for the revision of our article and all the comments.

Comment: The background provides a clear context for the study, emphasizing the importance of S-ICDs in preventing sudden cardiac death. However, it lacks some details about the significance of reproducibility and its impact on patient outcomes.

Authors’ response: We added the requested information to the text.

Comment: The materials and methods section provides a concise overview of the study design and data collection. However, it lacks information about the specific criteria for "acceptable vectors" and the methodology for device-based vector analysis. Adding more details about these aspects would enhance the transparency of the study.

Authors’ response: We expanded the methods to better explain all the details, as suggested.

Comment: Overall, the study addresses an important clinical issue and provides valuable insights. To enhance the paper, consider discussing the clinical implications of the findings in greater depth. Additionally, a brief discussion of future research directions would be beneficial.

Authors’ response: We added more comments regarding those issues, as requested.